# High-Pressure Delivery of Oncolytic Viruses via Needle-Free Injection Preserves Therapeutic Activity

**DOI:** 10.3390/cancers15235655

**Published:** 2023-11-30

**Authors:** Aida Said, Huy-Dung Hoang, Nathalie Earl, Xiao Xiang, Nadeem Siddiqui, Marceline Côté, Tommy Alain

**Affiliations:** 1Department of Biochemistry Microbiology and Immunology, University of Ottawa, Ottawa, ON K1H 8M5, Canada; asaid062@uottawa.ca (A.S.); hhoan085@uottawa.ca (H.-D.H.); xxian008@uottawa.ca (X.X.); marceline.cote@uottawa.ca (M.C.); 2Children’s Hospital of Eastern Ontario Research Institute, Ottawa, ON K1H 8L1, Canada; nearl@cheo.on.ca; 3Department of Biochemistry, McGill University, Montreal, QC H3G 1Y6, Canada; nadeem.siddiqui@mcgill.ca; 4Centre for Infection, Immunity and Inflammation, University of Ottawa, Ottawa, ON K1H 8M5, Canada

**Keywords:** needle-free injection, oncolytic virus, tumour, immunotherapy

## Abstract

**Simple Summary:**

We explored the use of needle-free injection (NFI) technology to deliver oncolytic viruses (OVs) for treating solid tumours. The study assesses the infectivity of OVs following NFI and compares the effectiveness of intratumoural administration of Vesicular Stomatitis Virus (VSV) to that of traditional needle injection (NI) in subcutaneous tumours in mice. NFI improves viral tissue distribution and preserves therapeutic activity of this virotherapy. The study establishes NFI as an efficient delivery method for OVs and underscores the need for further optimization to enhance therapeutic efficacy.

**Abstract:**

Intratumoural delivery of oncolytic viruses (OVs) to solid tumours is currently performed via multiple percutaneous methods of needle injections (NI). In this study, we investigated the potential use of a novel delivery approach, needle-free injection (NFI), to administer OVs to subcutaneous tumours. The stability and genetic integrity of several RNA and DNA viruses exposed to high-pressure jet injectors were first evaluated in vitro. We demonstrate that replication competence and infectivity of the viruses remained unchanged after NFI, as compared to traditional NI. Using the oncolytic Vesicular Stomatitis Virus expressing luciferase (VSVΔ51-Luc) in the syngeneic CT26 subcutaneous tumour model, we show that NFI administration not only successfully delivers infectious particles but also increases the dissemination of the virus within the tumour tissues when compared to NI. Furthermore, mice treated with VSVΔ51-Luc by NFI delivery showed similar reduction in tumour growth and survival compared to those with needle-administered virus. These results indicate that NFI represents a novel approach to administer and potentially increase the spread of OVs within accessible solid tumours, highlighting its usefulness in virotherapy.

## 1. Introduction

Oncolytic viruses (OVs) represent a promising immunotherapeutic approach to cancer treatment. OVs selectively replicate in cancer cells while sparing normal cells [1]. This selective infection can mediate tumour regression by directly lysing cancer cells, as well as exposing tumour-associated antigens and triggering host antitumour immune response [2,3,4]. However, several challenges must be overcome to maximize OV infections of solid tumours, including physical barriers, immune restrictions against the virus, immunosuppressive tumour microenvironment and tumour heterogeneity [3].

The initial hurdles in achieving sufficient infection lie in the physical barriers and elevated interstitial pressure of solid tumours. These factors restrict the distribution of oncolytic viruses (OVs) and contribute to suboptimal deliveries, diminishing the spread of OVs within the tumour tissue. Consequently, this limitation hampers the overall therapeutic efficacy of the treatment [5]. Intratumoural injection by needle, which elicits fewer side effects than other administration methods, is the most commonly used route of administration for superficial, accessible, solid tumours [6,7]. However, the extracellular matrix of solid tumours can often prevent the diffusion of OV-containing fluid throughout the tumour [5,8,9,10]. To circumvent this limitation, current intratumoural OV delivery uses non-traditional needles such as multipronged needles to increase fluid penetration [11], and combines chemical [12,13] or viral bioengineering [14,15] approaches, for instance, OVs expressing ECM-degrading enzymes [15,16,17], to help promote viral spread within the tumour tissues. Developing novel OV delivery methods that maximize initial viral spread is another avenue to consider for improving the therapeutic efficacy of oncolytic viral immunotherapies.

Needle-free injection (NFI) technology is a method to deliver medications without the need of a needle or syringe. In NFI, the therapeutic agent is delivered via a high-pressure, narrow jet stream of liquid that penetrates the skin and reaches into subcutaneous or intramuscular layers. The original purpose of NFI technology was to provide a more user-friendly and pain-free alternative to needle injection systems. This technology is currently being used to deliver effectively a variety of therapeutic agents, including insulin, other hormones, antibiotics, and vaccines; it has been shown to induce potent immunogenicity in the case of the vaccines [18,19,20,21,22]. NFI was also reported to result in better diffusion of the injected liquids compared to traditional needle injection in subcutaneous tumour [19,20], which could represent an added benefit for OV distribution.

In this study, we assessed the efficacy of first-generation NFI technology in delivering OVs to subcutaneous tumours. We first examined in vitro the replication competence of the oncolytic RNA viruses Vesicular Stomatitis Virus (VSVΔ51-GFP), Measles Virus (MeV-GFP) and Reovirus (ReoV), as well as the oncolytic DNA viruses Herpes Simplex Virus-1 (HSV1-GFP) and Vaccinia virus (VACV-GFP), when exposed to the high-pressure jet stream of needle-free injector system. All viruses subjected to NFI were found to infect and replicate to similar levels as compared to viruses delivered via traditional needle injection. We then investigated the intratumoural spread, as well as the effect on tumour growth and survival rate in vivo using a CT26 subcutaneous tumour- bearing mice administered VSVΔ51-Luc delivered either by NFI or NI. Improved luciferase expression, suggesting increased spread of virus, was detected in tumour tissues administered via NFI compared to that by NI. In addition, reduction in tumour growth and survival rate in mice treated with VSVΔ51-Luc delivered by NFI was found to be similar to those treated by NI. Overall, this study provides the first proof-of-principle for the use of NFI in delivering OVs, and our results demonstrate the potential of developing this delivery method for the treatment of solid tumours with OVs.

## 2. Materials and Methods

### 2.1. Cell Culture and Viruses

Murine colorectal carcinoma cell line CT26, human HEK293T, murine L929 and African green monkey Vero cells were obtained from the American Type Culture Collection (ATCC, Manassas, VA, USA). These cells were cultured in Dulbecco’s Modified Eagle Medium (DMEM, Grand Island, NY, USA) (Fisher, New York, NY, USA) supplemented with 10% fetal bovine serum (Sigma, Tokyo, Japan) and 0.1% penicillin and streptomycin (Life Technologies, Waltham, MA, USA) at 37 °C in 5% CO_2_.

HSV1 (HSV1-1716, strain 17–γ34.5 deleted, Sorrento Therapeutics, San Diego, CA, USA), VACV (JX-594 strain Wyeth, Tk-deleted expressing GM-CSF, Jennerex Biotherapeutics/Sillagen, Seoul, Republic of Korea) and ReoV (Type 3 Dearing, Oncolytics Biotech, Calgary, AB, Canada) were kindly provided by manufacturers. VSVΔ51-GFP and VSVΔ51-Luc (ΔM51 with insertion of GFP or Luc genes, respectively) were kindly provided by Dr. John Bell (Ottawa Hospital Research Institute). Vero cells were used to propagate HSV1, VACV, VSVΔ51-GFP and VSVΔ51-Luc viruses, and L929 cells were used to propagate ReoV. Briefly, cells were inoculated with viruses at the appropriate MOIs. Cells were then subjected to three freeze–thaw cycles to release intracellular viruses. Freeze–thawed lysates were clarified by centrifugation at 500× *g* for 10 min, and the supernatants were collected. Viral supernatants were subjected to ultracentrifugation at 28,000× *g* for 90 min on a sucrose cushion layer (36% sucrose, 10 mM HEPES, 150 mM NaCl, 0.1 mM EDTA, pH 7.3), with the exception of VSVΔ51-Luc. VSVΔ51-Luc for use in animal studies was purified using a 10–40% opti-prep gradient as previously described [23]. Titers of HSV1, VACV, VSVΔ51-GFP and VSVΔ51-Luc were determined by plaque titration using Vero cells monolayer and using 1% carboxymethylcellulose.

### 2.2. Western Blotting

For Western blot (WB) analysis to assess viral protein expression, cells were cultured in 6-well plates. After 12 h of infection, the cells were lysed in RIPA buffer (150 mM NaCl, 1.0% IGEPAL^®^ CA-630, 0.5% sodium deoxycholate, 0.1% SDS, 50 mM Tris, 50 mM NaF, 15 mM NaVO_3_, pH 8.0) supplemented with cOmplete Protease Inhibitor Cocktail (Roche, Basel, Switzerland). Cell debris were removed by centrifugation at 12,000 rpm for 5 min at 4 °C. The protein concentration was quantified using the DC Protein assay (BioRad, Hercules, CA, USA). The protein samples were loaded onto a 10% SDS-polyacrylamide gel and separated by electrophoresis. The separated proteins were then transferred to a PVDF membrane and blocked with Intercept (TBS) Blocking Buffer (LI-COR). Primary antibodies specific to HSV1 (Dako, #B011402), VACV (a kind gift of Dr. John Bell, Ottawa Hospital Research Institute), β-actin (#A5441, Sigma), and ReoV anti-serum (a kind gift of Dr. Earl Brown, University of Ottawa) were used. For secondary antibodies, IRDye^®^ 680RD Goat anti-Mouse IgG (LiCor, Lincoln, NE, USA, #926-68070) and IRDye^®^ 800CW Goat anti-Rabbit IgG Secondary Antibody (LiCor, #926-32211) (1:10,000) were used.

### 2.3. First Generation Needle-Free Device

The Inolife needle-free device (Inolife R&D Inc., NuGen Medical Devices, Toronto, ON, Canada) comes with injector reset box and sterile ampules (Figure 1A). To recharge the injector, the safety ring is moved to the “safe on” position, then the injector is placed in the reset box and the spring is fully loaded before closing the cover. When the injector is fully charged, the filled ampule is screwed into the injector, the safety ring is pulled to the “safe off” position. The injector is then placed on the subcutaneous tumour, and pressing the trigger releases the liquid contained inside the ampoule into the tissue via a high-pressure jet stream capable of tissue penetration. For in vitro virus stability assay, HSV1, VACV, VSV and ReoV were injected directly into a 6-well plate using either a one-end-hole needle or the Inolife needle-free device. The post-injection virus solutions then underwent three 10-fold dilutions before inoculated onto HEK293T cells (HSV1, VACV and VSV) or L929 cells (ReoV).

### 2.4. Immunofluorescence

To test for stability and virus replication of ReoV, the virus was inoculated onto L929 cells. After 48 h, the cells were fixed using 3.7% paraformaldehyde (PFA), then blocked for 30 min in blocking buffer (2% BSA + 0.1% Triton X-100). Cells were subsequently incubated with anti-ReoV anti-serum (Rabbit Ig, 1:1000) at room temperature for 30 min, then incubated with anti-rabbit Alexa Fluor 488 secondary antibody in blocking buffer (1:500, ThermoFisher, Waltham, MA, USA) at room temperature for 1 h. Immunofluorescent images was taking using EVOS cell imaging system (ThermoFisher).

### 2.5. Bromophenol Blue Injection and Detection Experiment

CT26 cells were subcutaneously injected, and when tumour volume reached approximately 1000 mm^3^, 50 μL of bromophenol blue dye was injected intratumourally either using the needle-free or needle injection method. Immediately after injection, the tumours were dissected to observe the distribution of the dye. Subsequently, images were captured to document the extent of dye accumulation within the tumour tissue.

### 2.6. Live Cell Monitoring of Virus Infection

To assess virus replication efficiency, HSV1, VACV, VSV and MeV were injected into a 6-well plate using either the needle injection or NFI method. The viruses were then inoculated onto HEK293T cells that were seeded in a 48-well plate. The cells were subsequently monitored for 72 h using the IncuCyte ZOOM^TM^ Live-Cell Analysis System (Sartorius, Göttingen, Germany). Images were captured every 2 h at 10× magnification to monitor viral infection of live cells based on fluorescence. The acquired images were analyzed using the accompanying IncuCyte ZOOM^TM^ (Incucyte S3 Software v2018A) (Sartorius). Measurement of total green object area (μm^2^/well) was utilized to determine the level of virus infection. Background subtraction was performed using the Top-Hat method with a disk-shaped structuring element containing a 10 mm radius in conjunction with a threshold of 1.0 green calibration unit. The analysis settings included edge split: Off, hole fill: No, adjust size: No.

### 2.7. CT26 Subcutaneous Tumour Model for In Vivo Imaging System (IVIS)

Female BALB/c mice, aged 5–6 weeks, were purchased from Charles River (Kingston, NY, USA). After one week of acclimatization, 5 × 10^5^ CT26 cells were subcutaneously injected into one flank of each mouse. After 14 days, the mice received intratumoural injections of 50 μL VSV-Luc at 10^8^ PFU/tumour using either the NI or NFI system. Twenty-four hours later, a solution of 15 mg/mL D-luciferin (GOLDBIO, CAT#: LUCK-1G) in DPBS was intraperitoneally injected at a dosage of 10 μL/g of body weight. Images were captured 5–8 min after luciferin injection. A series of 15 images, taken 1 min apart, with an acquisition time of 30 s per image, were obtained to determine the peak signal. The acquired images were analyzed using Living Image^®^ software V4.7 (for IVIS^®^ Spectrum images). The quantification of the region of interest was presented as the average radiance measurement, reported in units of photons per second per square centimeter per steradian (p/s/cm^2^/sr). The average radiance represents the average intensity of light emitted from the sample over a given area and time, providing quantitative information about the level of bioluminescent signal detected. Following the imaging procedure, the animals were sacrificed.

### 2.8. CT26 Subcutaneous Tumour Model

When tumours reached palpable size, approximately 5 × 5 mm, they were injected intratumourally twice on day 1 and day 3 with VSV-Luc at 1 × 10^8^ PFU/tumour. Tumour sizes were measured daily using a caliper, and their volumes were calculated using the formula [length × (width)^2^]/2 (ref). Animals were sacrificed either when an individual tumour reached 1500 mm^3^ or at an alternative humane endpoint. All animal procedures were conducted with approval from the University of Ottawa Animal Care Committee.

## 3. Results

### 3.1. High-Pressure Needle-Free Injection Does Not Affect Stability and Replication Competence of RNA and DNA OVs

We first conducted in vitro experiments to assess whether several clinically relevant OVs would be affected when subjected to high-pressure liquid jet injections as compared to needle injections (Figure 1A,B). MeV [24] and VSVΔ51 [25] are single-stranded, negative-sense, enveloped, non-segmented RNA viruses with a small genome; HSV1 [26] and VACV [27] are large, enveloped, linear, double-stranded DNA viruses; and ReoV is a segmented, non-enveloped, double-stranded RNA virus. To determine whether these viruses can retain therapeutic potential under the high-pressure conditions of NFI, the viruses were ejected into 6-well plates either through a traditional syringe and needle, or via the needle-free jet injector. The virus-containing solutions were then used to infect HEK293T cells as NFI directly onto monolayer of cells would have resulted in irreversible damage to the cells. To examine the ability of viruses exposed to NFI to infect and spread within the culture, the infections were monitored for 3 days under live cell imaging (Incucyte) using the tagged green fluorescent protein (GFP) (for HSV1, MeV, VACV, VSV) or by immunofluorescence (IF) (ReoV). We observed equivalent viral spread, as measured by immunofluorescence or GFP signal from Incucyte images, for cells infected by viruses subjected to needle injection versus NFI (Figure 2A,B). Additionally, viral protein expression was examined by Western blotting using antibodies raised against HSV1, VACV, VSVΔ51 or ReoV. Similarly, no differences were observed in total viral protein synthesis from infections resulting from viruses passed through needles or the needle-free jet injector (Figure 2C). Together, these data demonstrate that the infectivity of viruses is preserved after high pressure NFI exposure.

### 3.2. Intratumoural Delivery and Spread of VSVΔ51-Luc via Needle-Free Jet Injector

Needle-free injection technology was previously shown to deliver medicine deep into the skin, and even intramuscularly, in a spray-like and highly dispersed form [18,19,20]. Therefore, we anticipated that this method of delivery could be applied to subcutaneous, accessible tumours and provide a dispersed distribution that could help the viruses reach and infect more tumour cells. As a proof of concept, we used bromophenol blue dye for tracking liquid distribution after intratumoural injection by either needle or NFI. We found that NFI delivered a wider distribution of the dye within the tumour as compared to needle administration (Figure 3A). To determine whether a more dispersed spread of injected liquid would result in a better distribution of OVs within the tumour tissues, we tested the use of NFI technology for OV administration in a CT26 subcutaneous mouse tumour model. We used the oncolytic virus VSVΔ51-Luc for the ease of tracking viral replication in vivo via its luciferase transgene using In Vivo Imaging System (IVIS). After 14 days of CT26 subcutaneous implantation, 10^8^ PFU of VSVΔ51-Luc was administered intratumourally using a traditional needle or the needle-free jet injector system, and luciferase activity was measured 24 h post administration using IVIS (Figure 3B,C). The average radiance value obtained from IVIS imaging represents the average intensity of the emitted light across the region of interest, providing a quantitative measure of the bioluminescent signal emitted from the luciferase reporter gene. We found that NFI not only preserved OV replication, but also provided a significant increase in bioluminescence and intensity of VSVΔ51-Luc intratumourally (*p* < 0.0001) (Figure 3D,E). Notably, there was no significant difference in initial tumour volumes from the two mouse groups (*p* = 0.9375) (Figure 3F), suggesting the increase in luciferase activity appears to be a direct consequence of better virus distribution and infection rather than a variation in tumour sizes.

### 3.3. Therapeutic Efficacy of VSVΔ51-Luc Administered via Needle-Free Jet Injector

Given the promising effect of NFI on intratumoural spread of OV, we next tested whether this could translate into a better outcome in tumour volume reduction and survival. We repeated the subcutaneous CT26 tumour implantation model. When tumours reached 50–60 mm^3^, two intratumoural injections of 1 × 10^8^ PFU of VSVΔ51-Luc were administered on day one and three to maximize the antitumour effect of this OV (Figure 4A). As expected from previous experiences, NI of VSVΔ51-Luc resulted in a significant reduction in tumour volume, as well as a prolonged survival rate compared to the control (PBS) group (*p* < 0.0001 and *p* = 0.0042, respectively) (Figure 4B,C). However, NFI of VSVΔ51-Luc caused an equally significant reduction in tumour volume and improvement in survival rate when compared to the control group (*p* < 0.0001 and *p* = 0.0018, respectively). While in this model there was no significant difference in tumour volume and survival between the NI and NFI groups (*p* = 0.6164 and *p* = 0.6758, respectively), these results confirm that NFI of OVs can provide a similar performance in terms of therapeutic outcome for the treatment of subcutaneous solid tumours.

## 4. Discussion

The usage of needle-free injection technology is gaining attention as a promising technique for the delivery of small therapeutic molecules like insulin [28]. The use of NFI technology was also proven to be an effective method for transdermal delivery of exosomes [20], and was shown recently to be highly effective in delivery of more complex biologics such as vaccines or even mRNA-lipid complex. Until now, the largest macromolecules reported to be successfully delivered by NFI were mRNA-lipid nanoparticles [19]. Studies conducted thus far have demonstrated the dispersion and spray-like properties of NFI for vaccines and drugs. Furthermore, several lines of evidence indicate that this injection method can elicit robust immune responses upon vaccine administration. For example, when a needle-free injection system was used to administer the ZyCoV-D DNA or mRNA-based vaccine against SARS-CoV-2, significant antibody titers and immune responses were observed [19,29]. These responses included elevated levels of IgG and neutralizing antibodies, enhanced lymphocyte proliferation, and cytokine responses [19,29]. However, despite possessing favorable properties, NFI had yet to be tested for the delivery of OV. In this study, we are the first to evaluate the use of NFI for OV delivery and the effect of this administration method on OV efficacy in vivo. Using a collection of clinically important OVs, we showed that the high-pressure ejection by NFI does not compromise the replication competence of both RNA and DNA viruses. We also observed an improved distribution of VSVΔ51-Luc within tumour tissues using high-pressure needle-free administration in the subcutaneous CT26 tumour model. Finally, we confirmed that VSVΔ51-Luc injected by NFI into solid subcutaneous tumours provides equivalent tumour growth control and mouse survival to that provided by NI.

Intratumoural distribution and replication of different types of OVs plays a crucial role in stimulating effective antitumour immune responses [30]. OV infection induces immunogenic cell death, characterized by the release of damage-associated molecular patterns upon tumour cell death, which is essential for initiating immune reactions [31]. Compared to intravenous administration, intratumoural injection of OVs can concentrate the virus inside the tumour to facilitate induction of immunogenic cell death, better modulate the tumour microenvironment to overcome immunosuppression [32], and consequently enhance local and systemic antitumour immunity. Intratumoural delivery also overcomes potential unwanted side effects of systemic administration of OV [33,34]. Nonetheless, the biophysical properties of certain solid tumours present significant obstacles to effective delivery of intratumourally administered OVs [34,35]. Current injection methods are presently being improved for intratumoural delivery of drugs in general [36,37,38]. The conventional end-hole needles are the widely used method to deliver drugs or OVs intratumourally, at least in experimental laboratory settings. However, closed diamond tip multisided hole needles have been shown to be significantly more efficient at promoting drug spread intratumourally and at inducing antitumour immunity compared to one end-hole needles [36]. Additionally, three-side-hole needles or three-pronged array needles can demonstrate better dispersion and distribution compared to single-side-hole needles, producing more spherical distributions within tissues, especially for percutaneous therapies [38,39]. All of these alternative injection methods are aimed at addressing the limitations associated with conventional needle injections, such as leakage out of the tumour, as well as achieving widespread distribution of drugs within cancerous tissues. Our results presented here propose the use and development of needle-free jet injector as a novel alternative to administer viral immunotherapeutics to accessible tumours such as melanoma or head and neck cancer.

This study was performed using a first-generation needle-free jet injector device designed for human administration of insulin into skin tissues. An important limitation of this study was that the injector’s pressure and size were not ideal for administration to mouse models of cancer, requiring careful handling for successful injections. Even though the device was not optimized for mouse cancer models, we observed improved viral distribution into the tumour tissues by NFI as compared to NI. Improved initial viral infection could ultimately promote better tumour growth control; however, in this model, two doses of VSVΔ51-Luc already provided strong cancer regression, rendering the observed increased viral spread less impactful on tumour reduction compared to NI. Nonetheless, our results suggest that optimization of the technology could achieve useful outcomes for intratumoural delivery of OVs. Other oncolytic viruses with limited therapeutic efficacy due to poor viral spread, and additional hard-to-treat cancer models, could ultimately exhibit elevated benefits from administration by NFI. One potential improvement could be achieved via altering the nozzle geometry to optimize diffusion of OV-containing liquid inside solid tumour [40]. In addition, the injection volume and density of OV-containing vehicle could also be modified to adjust the penetration and diffusion of viruses inside tumour [41]. Finally, ejection pressure could be optimized for each tumour type, density, and size. In conclusion, our study establishes NFI as a novel and promising alternative for intratumoural delivery of viral immunotherapeutics, and future advancements in needle-free injector system technology will optimize this approach to further enhance its therapeutic potential.

## 5. Conclusions

Needle-Free Injection technology represents an effective alternative delivery method for OVs to target accessible solid tumours. Overcoming the physical barriers of tumour microenvironments is a critical challenge for virotherapy, and this initial investigation demonstrates that NFI maintains the infectivity and therapeutic potential of OVs. In a subcutaneous mouse tumour model, NFI could even enhance the intratumoural distribution of virus, showcasing its promise as an efficient and convenient delivery method compared to traditional needle injections. While the study was performed using a first-generation device not originally designed for animal experiments, future optimization of this technology has the potential to improve the effectiveness and therapeutic distribution of OVs within tumour tissues, making it a promising route for the intratumoural administration of oncolytic virotherapies.

## Figures and Tables

**Figure 1 cancers-15-05655-f001:**
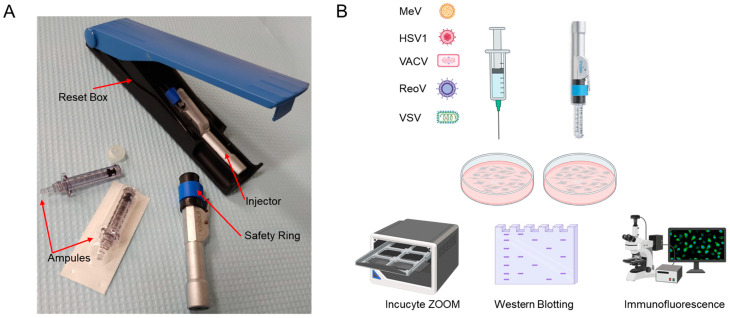
Experimental schema of oncolytic viruses subjected to needle or needle-free injector. (**A**): The first-generation Inojex Needle Free Injector System includes a reusable Inojex injector and reset box, as well as sterile, disposable ampules. (**B**): Two different delivery systems were utilized to infect HEK293T cells with five different oncolytic viruses, after which, viral infectivity was assessed. For MeV, HSV1, VACV and VSV, 72-h live-cell analysis was used to detect viral infection. For ReoV, immunofluorescence of viral protein after 24 h of infection was assayed. Additionally, for HSV1, VACV, VSV and ReoV, viral protein expression after 12 h of infection was analyzed via Western blotting.

**Figure 2 cancers-15-05655-f002:**
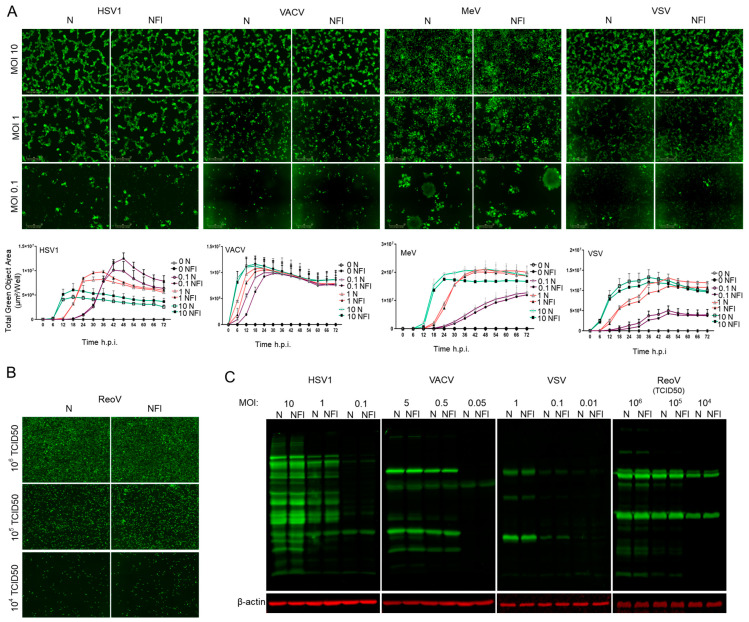
Needle-free injection system does not compromise virus infectivity in vitro. (**A**): Representative fluorescent images from Incucyte ZOOM^TM^ for MeV, HSV1, VACV, and VSV at 18 h post-infection from three different MOIs for each virus with respective quantifications below. Graphs depict the mean ± s.d. of four different images at 10X magnification taken every two hours for the total assay duration of 72 h. Data are representative of three independent experiments. (**B**): Representative immunofluorescent images of ReoV infectivity with 3 different titers (TCID_50_). 48 h after infection with ReoV, HEK293T cells were fixed and labeled using immunofluorescence. (**C**): Representative Western blots of viral protein from HEK293T cells 12 h after infection with HSV1, VACV, VSV or ReoV.

**Figure 3 cancers-15-05655-f003:**
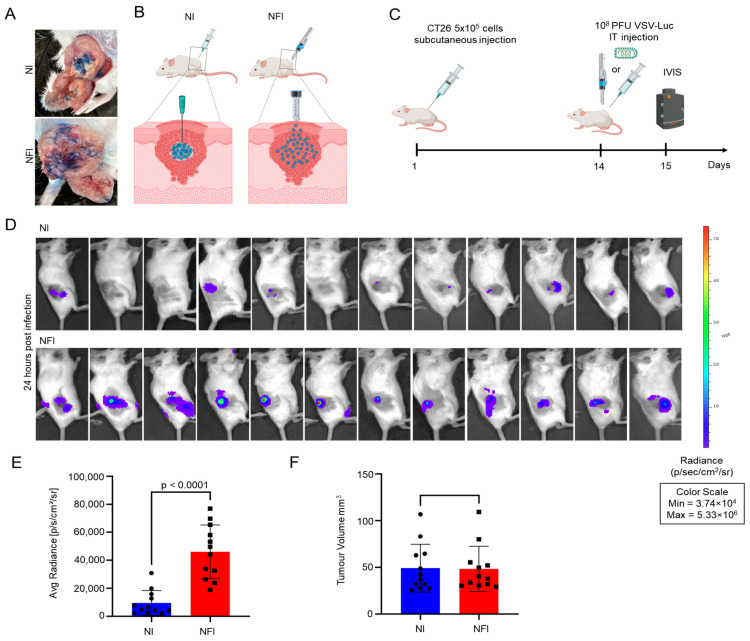
A needle-free injection system can efficiently deliver viruses to tumours in vivo. (**A**): Representative images for intratumoural delivery of bromophenol blue dye in large CT26 tumours using needle injection and needle-free injection. (**B**): Schematic representation of potential intratumoural viral distribution using a needle-free injection system. (**C**): Experiment timeline: 5 × 10^5^ cells were injected subcutaneously into the flanks of mice, then 14 days later mice were injected intratumourally with 1 × 10^8^ PFU of VSVΔ51-Luc. Twenty-four hours post-injection, mice were imaged to assess in vivo viral infection using IVIS imaging. (**D**): Representative bioluminescent images were taken using IVIS^®^ Spectrum imaging 24 h after virus delivery. n = 12 for each group. (**E**): Quantification of luminescence images using Living Image^®^ version 4.7 software. *p* < 0.0001 using a student’s *t*-test. (**F**): Tumour volume of mice at the time of VSVΔ51-Luc injection. *p* = 0.9375 using a student’s *t*-test.

**Figure 4 cancers-15-05655-f004:**
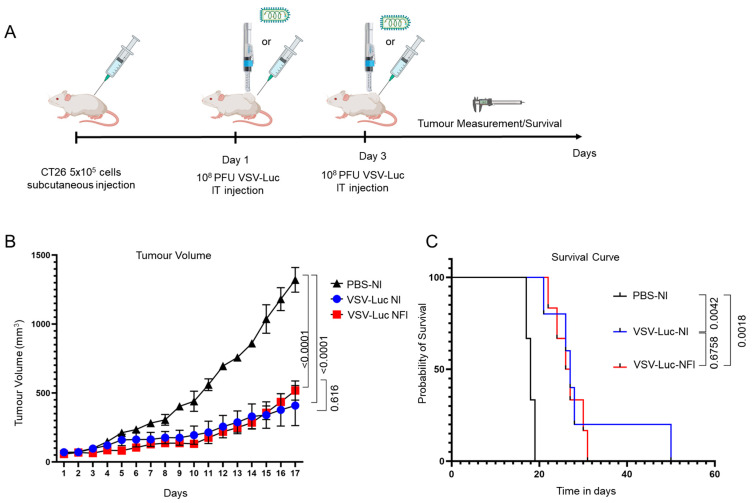
Needle-free injection of VSVΔ51-Luc is as effective for tumour treatment as needle-based injection. (**A**): Experiment outline: 5 × 10^5^ cells were injected subcutaneously in the right flank of mice. When tumours reached 50–60 mm^3^, a dose of 1 × 10^8^ PFU of VSVΔ51-Luc was injected intratumourally, followed by a second identical dose on day 3. Tumour volumes were measured every day. (**B**): Tumour volume curves showing the effect of a needle-free injection system on tumour growth (PBS-N: n = 3; VSV-Luc-N: n = 5; VSV-Luc-NFI: n = 6). Data are represented as arithmetic mean ± SEM. *p* < 0.05 using a two-way ANOVA with Tukey’s post-hoc test (**C**): Kaplan–Meier survival curve showing effects from the two different OV delivery systems (PBS-N: n = 3; VSV-Luc-N: n = 5; VSV-Luc-NFI: n = 6).

## Data Availability

The data presented in this study are available on request to the corresponding author.

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
