# Peer review of "High-Pressure Delivery of Oncolytic Viruses via Needle-Free Injection Preserves Therapeutic Activity"

_cancers, 2023, doi:10.3390/cancers15235655_

Round 1

Reviewer 1 Report

Comments and Suggestions for Authors

Said and colleagues compared the activity of oncolytic viruses after needle injection (NI) and needle-free injection (NFI) in vitro and in vivo. After injection of five oncolytic strains (VSV, measles virus, reovirus, HSV-1, Vaccinia) into a 6-well plate and subsequent inoculation of this fluid onto cell culture, GFP expression and viral protein expression was monitored via IncuCyte and Western blot, respectively. Viral spread was comparable between NI and NFI. 

After s.c. implantation of syngeneic murine colorectal carcinoma cells CT26 into the flanks of female BALB/c mice, bromophenol blue dye was distributed in the tumors more efficiently by NFI than by NI. A similar effect was observed for NFI compared to NI, when VSVΔ51-Luc was applied to the implanted tumors at day 14 and 16. In their final experiment, the authors compared the oncolytic activity of VSVΔ51-Luc and the survival of mice compared to control mice inoculated with PBS. NFI and NI of the oncolytic virus reduced the tumor volume similarly well and significantly better than PBS, concomitant with the survival of the mice. 

The application of oncolytic viruses via NFI is an important milestone, because it makes the application much easier. Therefore, the data of Said and colleagues represents significant progress. Some points remain to be answered: 

-         The authors argue that the improved spread of the oncolytic virus in the tumor by NFI will improve the outcome. However, despite the improved spread, the mice did not show improved oncolysis nor improved survival. How do the authors explain this discrepancy? This point should be discussed.

-        Have the authors tried to directly apply oncolytic viruses to the cell layers via NFI and NI? This experiment is important, because it will answer the question whether the monolayer of cells survives direct application of the jet stream. Additional experimentation is required.

-        L. 61-65: This sentence is included twice.

Author Response

Reviewer 1:

Said and colleagues compared the activity of oncolytic viruses after needle injection (NI) and needle-free injection (NFI) in vitro and in vivo. After injection of five oncolytic strains (VSV, measles virus, reovirus, HSV-1, Vaccinia) into a 6-well plate and subsequent inoculation of this fluid onto cell culture, GFP expression and viral protein expression was monitored via IncuCyte and Western blot, respectively. Viral spread was comparable between NI and NFI. 

After s.c. implantation of syngeneic murine colorectal carcinoma cells CT26 into the flanks of female BALB/c mice, bromophenol blue dye was distributed in the tumors more efficiently by NFI than by NI. A similar effect was observed for NFI compared to NI, when VSVΔ51-Luc was applied to the implanted tumors at day 14 and 16. In their final experiment, the authors compared the oncolytic activity of VSVΔ51-Luc and the survival of mice compared to control mice inoculated with PBS. NFI and NI of the oncolytic virus reduced the tumor volume similarly well and significantly better than PBS, concomitant with the survival of the mice. 

The application of oncolytic viruses via NFI is an important milestone, because it makes the application much easier. Therefore, the data of Said and colleagues represents significant progress. Some points remain to be answered: 

We thank the Reviewer for the positive and insightful comments.

  1. The authors argue that the improved spread of the oncolytic virus in the tumor by NFI will improve the outcome. However, despite the improved spread, the mice did not show improved oncolysis nor improved survival. How do the authors explain this discrepancy? This point should be discussed.

We agree with the Reviewer and have now added more discussion about this aspect within the revised manuscript. While we observed an elevated distribution of VSVΔ51-Luc within the tumour tissues, in this CT26 mouse model we found that two traditional needle injections of VSVΔ51-Luc on its own was also sufficient to induce significant tumour regression and prolonged survival. It is likely that in this specific mouse model, two doses of VSVΔ51-Luc, either through NI or NFI, could already reach the limit of therapeutic efficacy. Nonetheless, the proof-of-concept of using a needle-free injection system as a method to deliver intratumoural oncolytic viruses was demonstrated here. Importantly, other oncolytic viruses with limited therapeutic efficacy due to poor viral spread, and certain hard-to-treat cancer models, could benefit more from administration by NFI. We also emphasize in our revised manuscript that the needle-free injection device used in this study was a first-generation injector designed primarily for human applications such as insulin administration. The pressure of this device was certainly not optimal for murine tissue delivery, but now with the development of more sophisticated devices specifically tailored for mice or pets, with adjustable pressure settings to ensure precise intratumoural injections, future investigations into this technology will further confirm the therapeutic potential of intratumoural NFI for oncolytic virotherapy. We have added a paragraph mentioning these points in the Discussion section of the revised manuscript (lines 334-346).

  1. Have the authors tried to directly apply oncolytic viruses to the cell layers via NFI and NI? This experiment is important, because it will answer the question whether the monolayer of cells survives direct application of the jet stream. Additional experimentation is required.

The liquid jet system employed with needle-free injection operates at very high pressure to penetrate tissue, direct application to cell monolayers would have been extremely disruptive causing cell rupture, leading to their expulsion from the culture dish/plate, and ultimately compromising our viral spread assays in vitro when compared to needle injection. We have included an additional description for further clarification in the Results section of this revised manuscript (lines 202-204).

  1. L. 61-65: This sentence is included twice.

We thank the reviewer and apologise for this oversight during proof-reading. The duplicate sentence has now been removed in this revised version of the manuscript, thank you.

Reviewer 2 Report

Comments and Suggestions for Authors

In their study, Said et al. investigated the suitability of a needle-free injection (NFI) system for the application of various RNA and DNA viruses in oncolytic virotherapy in vitro and in vivo. They showed that the NFI system used works in vitro without affecting the activity of the viruses. In vivo, subcutaneous CT-26 tumors were injected with a luciferase-expressing VSV by conventional needle injection and by NFI. NFI injection allowed for better distribution and stronger infection of the tumor. The therapeutic efficiency, in terms of reduction of tumor growth and survival of the animals, was similar between NFI and the needle injection.

The study is very interesting, well performed and well written.

Minor points:

In which tumors was the bromophenol blue detection performed? The method used in Fig. 3A should also be described in the Materials and methods.

Author Response

Reviewer 2:

In their study, Said et al. investigated the suitability of a needle-free injection (NFI) system for the application of various RNA and DNA viruses in oncolytic virotherapy in vitro and in vivo. They showed that the NFI system used works in vitro without affecting the activity of the viruses. In vivo, subcutaneous CT-26 tumors were injected with a luciferase-expressing VSV by conventional needle injection and by NFI. NFI injection allowed for better distribution and stronger infection of the tumor. The therapeutic efficiency, in terms of reduction of tumor growth and survival of the animals, was similar between NFI and the needle injection.

The study is very interesting, well performed and well written.

We thank the reviewer for the positive comments.

Minor points: In which tumors was the bromophenol blue detection performed? The method used in Fig. 3A should also be described in the Materials and methods.

Thank you for pointing this out, this description has now been added in the Material and Methods section of this revised version of the manuscript (lines 150-155) and in the figure legend. The bromophenol blue detection experiment was also conducted in CT26 tumours as described below: CT26 cells were subcutaneously injected, and when tumour volume reached approximately 1000 mm3, 50 μl of bromophenol blue dye was injected intratumourally either using needle-free or needle injection method. Immediately after injection, tumours were dissected and imaged to observe the distribution of the dye.